

# Density regulation amplifies environmentally induced population fluctuations

Crispin M. Mutshinda[1], Aditya Mishra[2], Zoe V. Finkel[3] and Andrew J. Irwin[1]

[1] Department of Mathematics & Statistics, Dalhousie University, Halifax, NS, Canada
[2] Flatiron Institute, New York, NY, USA
[3] Department of Oceanography, Dalhousie University, Halifax, NS, Canada

Corresponding author
Crispin M. Mutshinda,
crispin.mutshinda@dal.ca

## ABSTRACT

**Background**. Density-dependent regulation is ubiquitous in population dynamics, and its potential interaction with environmental stochasticity complicates the characterization of the random component of population dynamics. Yet, this issue has not received attention commensurate with its relevance for descriptive and predictive modeling of population dynamics. Here we use a Bayesian modeling approach to investigate the contribution of density regulation to population variability in stochastic environments.
**Methods**. We analytically derive a formula linking the stationary variance of population abundance/density under Gompertz regulation in a stochastic environment with constant variance to the environmental variance and the strength of density feedback, to investigate whether and how density regulation affects the stationary variance. We examine through simulations whether the relationship between stationary variance and density regulation inferred analytically under the Gompertz model carries over to the Ricker model, widely used in population dynamics modeling.
**Results**. The analytical decomposition of the stationary variance under stochastic Gompertz dynamics implies higher variability for strongly regulated populations. Simulation results demonstrate that the pattern of increasing population variability with increasing density feedback found under the Gompertz model holds for the Ricker model as well, and is expected to be a general phenomenon with stochastic population models. We also analytically established and empirically validated that the square of the autoregressive parameter of the Gompertz model in AR(1) form represents the proportion of stationary variance due to density dependence.
**Discussion**. Our results suggest that neither environmental stochasticity nor density regulation can alone explain the patterns of population variability in stochastic environments, as these two components of temporal variation interact, with a tendency for density regulation to amplify the magnitude of environmentally induced population fluctuations. This finding has far-reaching implications for population viability. It implies that intense intra-specific resource competition increases the risk of environment-driven population collapse at high density, making opportune harvesting a sensible practice for improving the resistance of managed populations such as fish stocks to environmental perturbations. The separation of density-dependent and density-independent processes will help improve population dynamics modeling, while providing a basis for evaluating the relative importance of these two categories of processes that remains a topic of long-standing controversy among ecologists.

## INTRODUCTION

Density-dependent regulation is a pervasive feature of population growth processes. It often operates in concert with demographic stochasticity and environmental noise to generate temporal fluctuations in population abundance/density (*e.g.*, *May, Hassel & Southwood, 1974*; *Royama, 1992*; *Woiwod & Hanski, 1992*; *Lande, Engen & Saether, 2003*; *Brook & Bradshaw, 2006*). The prevailing form of density regulation is compensation or negative density dependence, which describes a pattern of decreasing population growth rate with increasing population abundance/density, and *vice-versa*. This phenomenon may result from intraspecific competition for limited resources (*e.g.*, *Hansen et al., 1999*) or other mechanisms such as predation and diseases that affect net population growth rate in a density-dependent fashion (*Hixon, Pacala & Sandin, 2002*).

*Hixon, Pacala & Sandin (2002)* highlight the following three salient features of populations undergoing compensatory density regulation. (1) Persistence: the population persists for many generations; (2) boundedness: the population size remains between some positive lower and upper bounds, and (3) return tendency: the population tends to increase below a certain threshold, the equilibrium level, and to decrease when above that threshold. The return tendency is a stabilizing mechanism due to its proclivity to restore populations to their equilibrium levels following a disturbance (*Murdoch, 1994*; *Yodzis, 1995*). This feature explains how the harvesting of an abundant population may increase rather than decrease total production in the next generation, and is consequently essential to the concept of sustainable yield in fisheries and wildlife management (*Rose et al., 2001*; *Fowler, 1987*).

A relatively less documented form of density dependence is depensation also known as inverse density dependence, positive density dependence or Allee effect (*Allee & Bowen, 1932*), which refers to a pattern of decreasing population growth with decreasing population density at low densities. This form of density dependence may arise from a variety of mechanisms, including the tendency for predators to kill a fixed number of prey, causing the death rate of the prey population to be higher at low density, and the decrease in birth rate at low population densities due to the difficulty of finding mates. Though rarely detected in practice, the Allee effect is reportedly widespread in nature (*Allee & Bowen, 1932*; *Dennis, 1989*; *Courchamp, Clutton-Brock & Grenfell, 1999*; *Kramer et al., 2009*). There are at least two reasons for the rare detection of the Allee phenomenon namely, (1) the difficulty of detecting natural populations at low density, and (2) the distortion of statistical analyses by the high variance inherent in small sample sizes (*Drake & Kramer, 2011*; *Kramer et al., 2009*). *Drake & Kramer (2011)* extended the logistic growth model to incorporate the Allee effects. Here we restrict our attention to negative density dependence.

There has been since the 1920s a long debate among ecologists on the relative importance of exogenous (environmental) *versus* endogenous (density-dependent) factors in driving

temporal fluctuations in population size/density. This debate reached its pinnacle during 1950s with on the one hand the density-independent and on the other hand, the density-dependent lines of thought led respectively by *Andrewartha & Birch (1954)* and *Nicholson (1957)*. The discovery, by *May (1976)* that simple discrete-time models of density dependence could generate very complex and potentially chaotic dynamics provided grist to the mills of Nicholson's school, reinforcing the view that observed empirical patterns could be explained without resorting to stochastic factors.

Over recent decades, evaluations of model predictions against observed patterns have resulted in a broad recognition that temporal fluctuations in population abundance/density result from both density-independent and density-dependent factors, potentially interacting in non-trivial ways (*Coulson, Rohani & Pascual, 2004*). This emerging consensus has shifted the research agenda in population ecology from the simple detection of density-dependence in population time series to the assessment of population dynamical consequences of density regulation. Despite its high relevance for descriptive and predictive population dynamics modeling, the interplay of density-dependent and density-independent population dynamics processes has not received adequate attention (but see *e.g.*, *Fromentin et al., 2001*; *Ohlberger, Rogers & Stenseth, 2014*).

In this study, we analyze, through a combination of analytical derivations and numerical simulations, the interaction between environmental stochasticity and density regulation in driving temporal fluctuations in population abundance. Using the stochastic Gompertz model to describe the population dynamics in a stochastic environment with constant variance, we analytically derive an explicit formula linking the stationary variance of population abundance/density to the environmental variance and the strength of density feedback. We derive a formula quantifying the contribution of density regulation to population variance in stationary phase. We conduct a simulation study to corroborate empirically the analytically established relationship between density feedback and stationary variance and to check whether the same pattern of association holds for the Ricker model (*Ricker, 1954*), which is also widely used for population dynamics modeling, particularly in fisheries.

## MATERIALS & METHODS

### Model specification and analytical derivations

Let $Y_0$ and $Y_t$ denote respectively the initial population size (population size at time 0) and the population size at time $t\,(t \geq 1)$. We assume a discrete-time stochastic Gompertz model for the population dynamics (*Reddingius, 1971*; *Royama, 1981*; *Sibly et al., 2005*; *Dennis et al., 2006*; *Mutshinda & O'Hara, 2011*; *Mutshinda, O'Hara & Woiwod, 2009*; *Mutshinda, O'Hara & Woiwod, 2011*; *Mutshinda et al., 2019*) so that the population size at time $t \geq 1$ is given by

$$Y_t = Y_{t-1} \exp \left\{ r \left( 1 - \frac{\ln(Y_{t-1})}{k} \right) + \varepsilon_t \right\} \tag{1}$$

where $r > 0$ is the intrinsic growth rate (*i.e.*, $e^r$ is the multiplicative population growth rate in the absence of density dependence), whereas the realized growth rate $Y_t / Y_{t-1}$

decreases with increasing population size $Y_{t-1}$, and $k > 0$ is the natural logarithm of the carrying capacity or equilibrium population size/density denoted by $K$. The error term $\varepsilon_t$, assumed to be Gaussian with mean zero and variance $\sigma_t^2$, typically integrates process stochasticity (demographic and environmental stochasticity) and observation error. The Gompertz model as presented here is phenomenological and only describes population level changes without explicitly accounting for individual level processes or age structure, in contrast with mechanistic models that translate individual parameters into population level consequences.

For the purpose of the present study, we rely on simulated data and so, we assume that the population size is observed without error. In addition, we know the data-generating model when in practice the error term typically involves the effect of model misspecification as well. For small populations, demographic stochasticity may be important, and its inverse scaling with the population size introduces time-dependence in the random error term $\sigma_t^2$ (*e.g.*, *Saether et al., 2000*). Since our focus is on the population dynamics in stationary phase, we assume that the population size is large enough for demographic stochasticity to be unimportant. Therefore, we consider the process error to be entirely due to environmental fluctuations in a stochastic environment with constant variance, so that $\sigma_t^2 = \sigma^2$ with the error terms $\varepsilon_t$, $t \geq 1$ being serially independent. On the natural logarithmic scale with $y_t = \ln(Y_t)$, model (1) becomes

$$y_t = y_{t-1} + r\left(1 - \frac{y_{t-1}}{k}\right) + \varepsilon_t. \tag{2}$$

Letting $\beta = 1 - rk^{-1}$, we can re-write model (2) as

$$y_t = r + \beta y_{t-1} + \varepsilon_t \tag{3}$$

which is a first-order autoregressive [AR(1)] model with autoregressive coefficient $\beta$.

From the expression of $\beta$ in terms of $r$ and $k$, it follows that $k = r(1 - \beta)^{-1}$, which establishes a one-to-one correspondence between the parameters of the Gompertz model in AR(1) form (Eq. 3) and those of the standard Gompertz model (Eq. 1). However, the AR(1) form is more convenient for analyzing the model's dynamic behavior as we do next.

We start by analyzing the dynamic behavior of the deterministic Gompertz model in AR(1) form (*i.e.*, Eq. 3 without the error term $\varepsilon_t$). We discuss the existence of an equilibrium point and the pace at which the population size/density approaches the equilibrium. We then consider the stochastic version of the model (Eq. 3) for which the equilibrium is a non-degenerate distribution (the stationary distribution) rather than being a single point as in the deterministic model. We analyze the temporal fluctuations of the population size around the mean value of its stationary distribution.

## Dynamic behavior of the deterministic Gompertz model

The deterministic part of the Gompertz model in AR(1) form is given by the first-order difference equation

$$y_t = r + \beta y_{t-1}. \tag{4}$$

If $|\beta| < 1$, the system will eventually converge to an equilibrium state $y_\infty$ at which the population size remains constant, meaning that $y_\infty = r + \beta y_\infty$. Solving this equation for $y_\infty$ yields the following expression for the non-trivial equilibrium

$$y_\infty = r(1-\beta)^{-1} = k. \tag{5}$$

We can use the relation $r = y_\infty(1-\beta)$ resulting from (5) to describe the population dynamics in terms of $y_\infty$ as $y_1 = y_\infty + \beta(y_0 - y_\infty), y_2 = y_\infty + \beta^2(y_0 - y_\infty)$, and by induction we obtain the following closed-form solution to the first-order difference Eq. (4).

$$y_t = y_\infty + \beta^t (y_0 - y_\infty). \tag{6}$$

Consequently, $y_t$ will converge to $y_\infty$ provided that $|\beta| < 1$, with faster convergence when $|\beta|$ is closer to zero, with $y_t$ approaching the equilibrium state $y_\infty$ either monotonically from above or from below when $0 < \beta < 1$ or through damped oscillations when $-1 < \beta < 0$.

Following from Eq. (6), $y_t$ relates to $y_{t-1}$ as $y_t = y_\infty + \beta(y_{t-1} - y_\infty)$, which can be re-arranged as $y_t = (1-\beta)y_\infty + \beta y_{t-1}$. The relationship $y_t = (1-\beta)y_\infty + \beta y_{t-1}$ indicates that $y_t$ is a linear combination of $y_\infty$ and $y_{t-1}$ with the autoregressive coefficient $\beta$ providing a direct measure of the dependence of $y_t$ on $y_{t-1}$. As a result, the autoregressive parameter of the Gompertz model in AR(1) form is often referred to as the strength of density dependence (*e.g.*, *Hampton et al., 2013*; *Ponciano, Taper & Dennis, 2018*; *Messmer, 2019*; *Peeters et al., 2022*). We also adopt this terminology.

Deterministic models of population dynamics draw on the assumption that vital rates are constant over time, so that a single set of input values uniquely determines the output value. However, an obvious feature of the real world is that the environment varies continually. Environmental stochasticity refers to the variability in demographic rates caused by random variations in environmental conditions, whereas sampling variations in independent outcomes of demographic events (births, death and dispersal) among individuals in a finite population produces a different kind of fluctuations in population dynamics known as demographic stochasticity. While environmental stochasticity affects all populations irrespective of size, demographic stochasticity is generally only relevant in small populations due to its inverse scaling with the population size. We refer to *Lande, Engen & Saether (2003)* for details on statistical methods for estimating demographic and environmental stochasticity from empirical data. We next consider the dynamic behavior of the stochastic Gompertz model and investigate the interaction between density-dependent effects and environmental noise in driving temporal fluctuations in population abundance/density.

## Dynamic behavior of the stochastic Gompertz model

The stochastic Gompertz model in AR(1) form is given by Eq. (3). If $|\beta| < 1$ the population size will eventually reach a stationary distribution (or equilibrium distribution), which is the stochastic version of an equilibrium in the deterministic model. In stationary phase, the population growth rate is zero on average. Since the additive error $\varepsilon_t$ are assumed to be normally distributed, the stationary distribution is also normal with long-run mean value $\mu_\infty$ and variance $v_\infty$ determined by their time-invariance property. Taking expectations

of both sides of Eq. (3) and applying the time-invariance property of the long-run mean yields the equation $\mu_\infty = r + \beta\mu_\infty$ whose solution for $\mu_\infty$ is

$$\mu_\infty = r(1-\beta)^{-1} = k \tag{7}$$

Alternatively, taking variances of both sides of (3) and applying the time-invariance property of the long-term variance yields the equation $v_\infty = \beta^2 v_\infty + \sigma^2$ whose solution for $v_\infty$ is

$$v_\infty = \sigma^2 / (1 - \beta^2). \tag{8}$$

Equations (7) and (8) indicate that both the mean of the stationary distribution, which is identical to the deterministic stable equilibrium value $y_\infty = k$ (the log-carrying capacity), and the stationary variance (*i.e.*, the variance of population time series in stationary phase) depend on the strength, $\beta$, of density regulation. In addition, the stationary variance $v_\infty$ is proportional to the environmental variance, $\sigma^2$, and relates to the density feedback in such a way that stronger density regulation (*i.e.*, $|\beta|$ values close to 1) induces higher variability.

It is worth emphasizing that stationarity is a key requirement for analyzing density dependence in population time series, and Eqs. (7) and (8) only make sense for stationary time series. Therefore, the sensible range for the autoregressive parameter $\beta$ is the interval $(-1, 1)$. However, since we are interested in the stationary variance which depends on $\beta$ only through $\beta^2$, we restrict attention on $\beta$ values in the unit interval $0 \leq \beta < 1$. From our model assumptions that $r$ and $k$ are both positive and the expression $\beta = 1 - rk^{-1}$ connecting the autoregressive coefficient of the Gompertz model in AR(1) form to the parameters $r$ and $k$ of the standard Gompertz model, it follows that $0 \leq \beta < 1$ corresponds to $r \leq k$.

When $\beta = 0$, which corresponds to the case $r = k$ in the standard Gompertz model formulation, the population trajectory follows a Gaussian white noise process shifted at $r$. This process is stationary with constant variance $\sigma^2$. When $\beta = 1$, which arises as a limiting case when $k$ tends to infinity in the standard Gompertz model formulation, the dynamics are density independent, and the population trajectory is a random walk process with drift $r$. This process is not stationary since $\text{Var}(y_t) = t\sigma^2$ is unbounded and $\text{E}(y_t) = y_0 + rt$ is only constant when $r = 0$ (*i.e.*, for the random walk without drift), while a key condition for (weak) stationarity is that the mean and variance of the time series are constant, the other requirement being that for any integer $h$, $\text{Cov}(y_t, y_{t+h})$ depends only on $h$ and not on $t$.

For density-dependent dynamics where $\beta$ is statistically different from zero and $|\beta| < 1$, the population fluctuates around the stationary mean $\mu_\infty$ with variance $\sigma^2/(1-\beta^2)$, where $\sigma^2$ represents the environmental variance. Therefore, the portion of the stationary variance due to density regulation, herein denoted by $\sigma_{dd}^2$, is given by the difference between the stationary variance $\sigma^2/(1-\beta^2)$ under density-regulation and its counterpart $\sigma^2$ in the absence of density feedback on population growth rate. That is,

$$\sigma_{dd}^2 = \beta^2 \sigma^2 / (1 - \beta^2). \tag{9}$$

Consequently, the proportion of the stationary variance $v_\infty$ due to density regulation is

$$\varphi_{dd} = \sigma_{dd}^2 / v_\infty = \beta^2. \tag{10}$$

Interestingly, the proportion of the stationary variance $v_\infty$ due to density regulation is simply the square of the density-dependence effect $\beta$. This provides a biological interpretation of the autoregressive coefficient of the AR(1) model when applied to population dynamics, besides being a measure of the strength of density dependence. Equation (9) provides a tool for separating the stationary variance of population time series into a density-independent component and a density dependent counterpart.

A key motivation for using the Gompertz model is that it can be written as an AR(1) model, which is simple with known statistical properties and established inferential procedures (*Mutshinda & O'Hara, 2010*). Because the AR(1) model is a linear model with Gaussian errors, maximum likelihood estimates of $r$, $\beta$, and $\sigma^2$, which are identical to least square estimates, can be obtained by performing a linear regression of $y_t$ on $y_{t-1}$ ($t = 1, 2, 3, \ldots$) using standard statistical packages. However, confidence intervals from standard statistical packages are not valid because the $y_t's$ are not independent due to the autoregressive structure in the model. *Dennis & Taper (1994)* discuss bootstrap and jackknife procedures for constructing confidence intervals of the AR parameters. In a Bayesian framework, priors distributions can be defined to appropriately constrain the model parameters to more sensible range of values, thereby reducing posterior uncertainty while guaranteeing parameter identifiability (*Mutshinda, 2009a*; *Mwanza, 2010*).

A common objection to the Gompertz model is that the growth rate depends, if applicable, only logarithmically on the population density allegedly inducing weaker density feedback than the closely related Ricker model (*Ricker, 1954*) where the growth rate at time $t$ depends on $Y_{t-1}$, the actual population size (*Dennis & Taper, 1994*). The Stochastic Ricker model is given by

$$Y_t = Y_{t-1} \exp\left\{ r\left(1 - \frac{Y_{t-1}}{K}\right) + \varepsilon_t \right\}, \tag{11}$$

where $r$ is the intrinsic growth rate and $K$ is the carrying capacity or equilibrium population density on the scale of untransformed population size, and $\varepsilon_t$ are zero-mean random shocks to the population growth rate assumed to be normally distributed and serially independent. The Ricker model is widely used for population dynamics modeling, particularly in fisheries.

In the next section, we describe a simulation study designed to corroborate empirically the pattern of increasing stationary variance with increasing strength of density regulation under the Gompertz model and to investigate whether this pattern carries over to the Ricker model.

## Report on the simulation study
### Data simulation
Given numerical values of the parameters $r, k$ and $\sigma$, an initial log population size $y_0$, and a routine for generating standard normal random variables, one can recursively generate trajectories $y_0, y_1, y_2, \ldots y_n$ from the Gompertz model with initial population density set to the log-carrying capacity. We simulated replicated population time series over 300 time

steps and discarded the first 200 observations to ensure that the last 100 data points come from the stationary distribution. We tuned the parameters of the data-simulating model to mimic different levels of density regulation and different levels of environmental noise. Since the parameter $\beta$ only depends on the positive parameters $r$ and $k$ through the ratio $r/k$, one can generate data with different levels of density-regulation (different $\beta$ values) by fixing one of the two parameters, typically the carrying capacity, and varying the other (*e.g.*, *Greenwell & Ng, 1984*). Fixing the carrying capacity to 1 amounts to expressing population density in units of the carrying capacity. In our simulations, we fix the log-carrying capacity to 1, so that $\beta = (1 - r)$ for $0 < r < 1$. We simulated data under four different levels of environmental noise, with environmental variance set to 0.10, 0.15, 0.20 and 0.25, and three different values of $r$ namely, 0.8, 0.6, and 0.4, corresponding to $\beta$ values 0.2, 0.4, and 0.6, respectively.

We graphically checked whether the empirical stationary variance $v_\infty$ increased with increasing the strength of density feedback as implied by the analytical result ((8)). We computed the contribution $\sigma_{dd}^2$ of density regulation to the empirical stationary variance $v_\infty$ at different levels of density regulation as the difference between the empirical stationary variance of simulated population trajectories and the assumed environmental variance.

In practice, the model parameters are unknown, and one has to rely on estimates obtained from the model fitting to data. We examined the relationship between the strength of density regulation and the proportion of stationary variance due to density regulation using environmental variance estimates over 100 replicated population trajectories at each of the three levels of density regulation under consideration.

A question that springs to mind is whether the pattern of increasing stationary variance with increasing strength of density feedback inferred under the Gompertz model holds for the Ricker model as well. Since under the Ricker model, unlike the Gompertz model, we do not have an expression separating the stationary variance into contributions from environmental stochasticity and density regulation, we rely on simulations to tackle this question. We fit, with a Bayesian approach (*Gelman et al., 2013*; *Mutshinda et al., 2022*), the stochastic Gompertz and stochastic Ricker models to data replicates simulated from the stochastic Gompertz model with different levels of density regulation. For the Gompertz model, we independently assigned a *Gamma*(1, 1) prior on the intrinsic growth rate r, a standard normal prior independently on the autoregressive coefficient $\beta$ and an InvGamma(0.1, 0.1) prior on the environmental variance $\sigma^2$. For the Ricker model, we independently assigned a *Gamma*(1, 1) prior on the on the intrinsic growth rate r and *Gamma*(0.1, 0.1) priors on the carrying capacity $K$ and the environmental variance $\sigma^2$. We used Markov chain Monte Carlo methods (*Gilks, Richardson & Spiegelhalter, 1996*) to simulate, *via* the Bayesian freeware OpenBUGS (*Thomas et al., 2006*; *Mutshinda, 2009b*), from the joint posterior distributions. We primarily ran the models in OpenBUGS to assess their convergence both informally through visual inspection of traceplots and autocorrelation plots in OpenBUGS, and formally by looking at the Gelman, Rubin statistics *via* the R package CODA (*Plummer et al., 2006*), and found that at least 2,000 iterations were required for convergence. Therefore, we used a burn-in period of 4,000 iterations for both models.

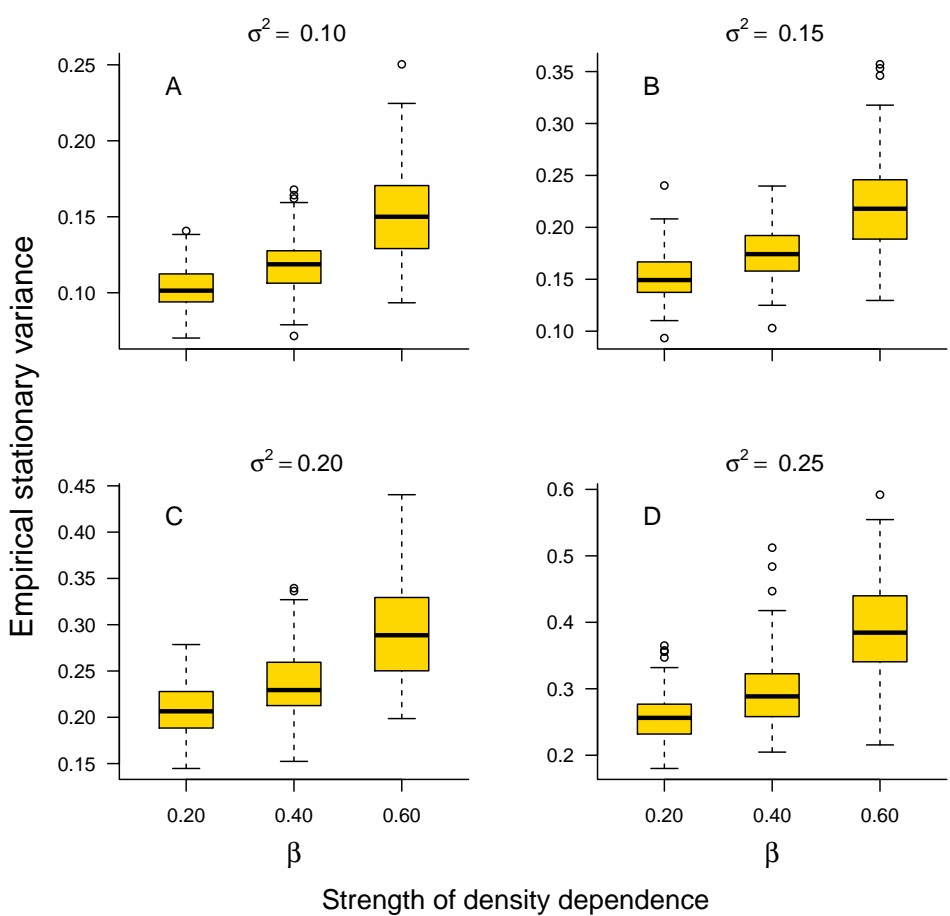

**Figure 1  Relationship between stationary variance and the strength of density dependence.** Box-and-whisker plots summarizing the distributions of the stationary variance of simulated population trajectories at different levels of density regulation, showing the monotonic increase of the stationary variance with the strength of density dependence in simulated population trajectories. For each boxplot, the height of the box indicates the 25th (Q1) and the 75th (Q3) per-centiles; the horizontal line inside the box is the median, and the lower and upper whisker limits are defined as $Q1 - 1.5 \times IQR$ and $Q3 + 1.5 \times IQR$, respectively, where IQR represents the interquartile range ($IQR = Q3 - Q1$). The dots placed beyond the whiskers' edges indicate outliers.

We estimated the contribution $\sigma_{dd}^2$ of density dependence to the stationary variance $v_\infty$ by the difference between the empirical stationary variance of simulated population time series and the posterior estimate of the environmental variance, and subsequently derived the proportion $\varphi_{dd}$ of stationary variance due to density regulation as $\varphi_{dd} = \sigma_{dd}^2 / v_\infty$.

## RESULTS

The stationary variance of simulated population trajectories increased monotonically with the strength of density regulation (Fig. 1).

This result indicates that in a given environment, strongly regulated populations will exhibit larger temporal fluctuations than weakly regulated ones. In addition, the proportion

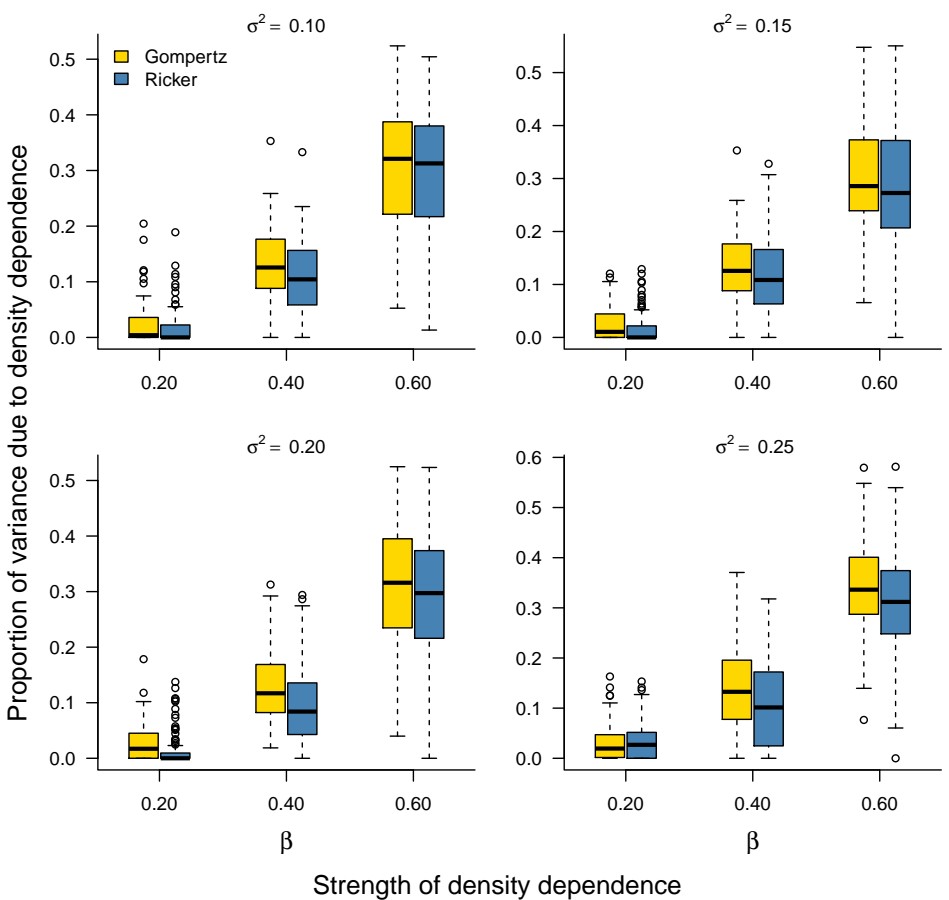

**Figure 2 Increasing proportion of stationary variance due density regulation with increasing strength of density dependence.** Box-and-whisker plots summarizing the distributions of the proportion of stationary variance (*i.e.,* the variance of population time series in stationary phase) due to density dependence under the stochastic Gompertz (gold fill) and Ricker (steel blue fill) models at different levels of density regulation.

$1 - \sigma^2/v_\infty$ of stationary variance due to density regulation increased monotonically with the strength of density dependence in simulated population time series (Fig. S1 in Online Supplemental Material).

As noted earlier, the environmental variance $\sigma^2$ is unknown in practice and needs to be estimated from data. Our Bayesian model fitting was effective at retrieving the environmental variance in population time series under both the stochastic Gompertz and stochastic Ricker models (Fig. S2). This allowed us to evaluate the proportion $\varphi_{dd} = \sigma^2_{dd}/v_\infty$ of stationary variance due to density regulation through Eq. (9). Under either model, the proportion of stationary variance due to density regulation increased monotonically with the strength of density feedback in the data (Fig. 2).

In addition, the inferred proportion of stationary variance due to density dependence was approximately equal to the square of the autoregressive parameter $\beta$ over simulated

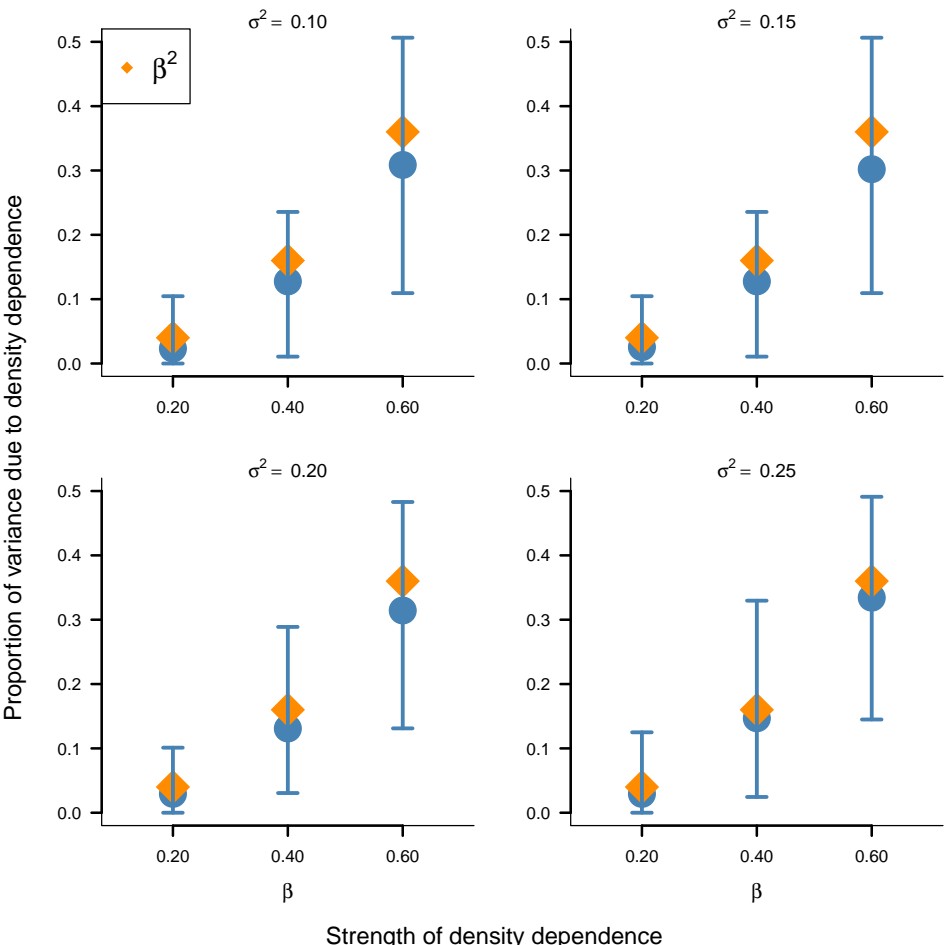

**Figure 3** **Relationship between the strength of density dependence and the proportion of stationary variance due to density dependence under the stochastic Gompertz model in AR(1) form.** Mean values (solid blue circles) and 95% confidence intervals of the proportion of stationary variance due to density regulation from fitted stochastic Gompertz model in AR(1) form over 100 synthetic population trajectories at different levels of environmental noise (environmental variance shown in each panel) against the strength of density regulation represented by the autoregressive coefficient. The overlaid orange diamonds indicate the square of the autoregressive coefficient in simulated data.

data replicates for the stochastic Gompertz model (Fig. 3), consistent with our analytical derivation in Eq. (10).

This result provides another biological meaning to the autoregressive coefficient of the AR(1) model when applied to population dynamics.

## DISCUSSION

In this study, we combined analytical derivations and numerical simulations to analyze the interplay between environmental noise and density regulation in driving temporal fluctuations in population abundance/density. We derived a formula (Eq. 8) relating the stationary variance of the population abundance/density under Gompertz-type density

regulation in a stochastic environment with constant variance to the environmental variance and the strength of density dependence, implying that density regulation amplifies the magnitude of environmentally induced population fluctuations. We worked out a formula separating the stationary variance in population abundance/density into its density-independent and density dependent components. An important result emerging from this variance decomposition is that the square of the autoregressive coefficient of the Gompertz model in AR(1) form represents the proportion of stationary variance due to density regulation (Eq. 10).

Simulation results substantiated empirically the analytically established pattern of increasing stationary variance with increasing strength of density regulation under the Gompertz model (Fig. 1). The Bayesian model fitting was effective at retrieving the environmental component of the stationary variance under both the stochastic Gompertz and stochastic Ricker models (Fig. S1). This allowed us to estimate the portion of stationary variance due to density-regulation by the difference between the stationary variance of simulated population trajectories and the posterior estimate of the environmental variance under either model.

We expect the pattern of increasing stationary variance with the strength of density feedback established analytically and/or empirically under the stochastic Gompertz and Ricker models (Fig. 2) implying higher temporal variability for strongly regulated populations to be a general phenomenon with population dynamical models. This is because the linear relationships in the Gompertz and Ricker models can be considered as Taylor approximations near equilibrium of more complex density-dependent growth functions (*Dennis & Constantino, 1988*).

Simulation results also substantiated empirically the analytically established result in Eq. (10) that the square of the autoregressive coefficient of the Gompertz model in AR(1) represents the proportion of stationary variance due to density regulation (Fig. 3). This finding provides another biological meaning to the autoregressive coefficient $\beta$ of the Gompertz model in AR(1) form: Besides being a measure of the strength of density dependence, $\beta$ represents the proportion of stationary variance due to density dependence.

Our main finding that density regulation amplifies environmentally induced population fluctuations has important implications for population viability. It suggests that intense intra-specific resource competition increases the risk of environment-driven population collapse at high density, lending support to opportune harvesting as a means to improve the resistance of managed populations such as fish stocks to environmental perturbations.

Overall, our analytical and empirical analyses demonstrate that the impact of density regulation on population dynamics involves a deterministic component affecting the mean population abundance/density and a stochastic component affecting the process variance. Therefore, the de facto decomposition of process variance into environmental stochasticity and demographic stochasticity disregards the contribution of density dependence to population variance in stochastic environments, which may be substantial.

## CONCLUSIONS

We analytically established and empirically verified that environmental noise interacts with density feedback in convoluted ways, causing density-regulated populations to undergo stronger fluctuations than expected under the sole influence of environmental stochasticity. The separation of exogenous (environmental) and endogenous (density-dependent) components of population variability is essential to evaluating their relative importance, which remains a topic of continuous debate among ecologists (*Coulson, Rohani & Pascual, 2004*).

Our analyses demonstrate that one can effectively extract the environmental component of temporal variability in population dynamics variability by fitting ecological models to population time series, and the Bayesian approach adopted here has proven fruitful to this end.

In order to decompose the stationary variance of population size in its density-independent component and density-dependent components, we made the following two simplifying assumptions: (1) Demographic stochasticity is unimportant, and (2) population sizes are recorded without error. While the irrelevance of demographic stochasticity in large populations is a sensible assumption due to the inverse scaling of demographic stochasticity with the population size (*Lande, Engen & Saether, 2003*), real-world observational population time series are typically fraught with observation or sampling errors. Inadequate handling of sampling error can induce wrong inferences about population processes, including spurious density dependence detection (*Dennis & Taper, 1994*; *Freckleton et al., 2006*). Density-dependent population dynamics models that integrate process variation and observation error are required to analyze the interaction of environmental stochasticity and density regulation in real-world populations. These kinds of models can be conveniently developed and fitted in the Bayesian state-space modelling framework (*e.g.*, *De Valpine & Hastings, 2002*; *Buckland et al., 2004*; *Dennis et al., 2006*).

### Funding

This work was supported by the Simons Collaboration on Computational Biogeochemical Modeling of Marine Ecosystems (CBIOMES) (Grant ID: 549935). The funders had no role in study design, data collection and analysis, decision to publish, or preparation of the manuscript.

### Grant Disclosures

The following grant information was disclosed by the authors:
Simons Collaboration on Computational Biogeochemical Modeling of Marine Ecosystems (CBIOMES): 549935.

### Competing Interests

The authors declare there are no competing interests.

## Author Contributions

- Crispin M. Mutshinda conceived and designed the experiments, performed the experiments, analyzed the data, authored or reviewed drafts of the article, and approved the final draft.
- Aditya Mishra analyzed the data, prepared figures and/or tables, authored or reviewed drafts of the article, and approved the final draft.
- Zoe V. Finkel conceived and designed the experiments, authored or reviewed drafts of the article, and approved the final draft.
- Andrew J. Irwin conceived and designed the experiments, performed the experiments, authored or reviewed drafts of the article, and approved the final draft.

## Data Availability

The R code for data simulation and the OpenBUGS code used to fit the stochastic Gompertz and stochastic Ricker models to simulated data replicates are available at GitHub: https://github.com/mutshinda/Density-Dependence/blob/main/Rcode_DensDep.R.

## Supplemental Information

Supplemental information for this article can be found online at http://dx.doi.org/10.7717/peerj.14701#supplemental-information.

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
