# Peer review of "Density regulation amplifies environmentally induced population fluctuations"

_PeerJ, doi:10.7717/peerj.14701_

## Round 0.1 · original submission · Major Revisions

I have now received three reviews of your manuscript - they all are quite positive and point out the paper paper is interesting and well written. There's, however, room for improvement. On one hand - several comments points give very helpful feedback on how to improve the presentation of your model results. On the other - reviewers raise valid issues about (i) biological reality of parameters used and (ii) adequacy of the models applied (given their simplistic assumptions). These points should be addressed to place the study in a wider context of more recent modelling approaches and to demonstrate the robustness of its outcomes.

Reviewer 1 ·

Basic reporting

Writing style is concise and unambiguous. There are few unnecessary repetitions, which could be omitted without the loss of flow (see 4). I have no further comments on language. Background is well presented so the the reader knows what is done and what for and – I believe – the literature is relevant. The structure is correct, but improvements could include dividing M&M into two main parts. I also have suggestions on how figures could be improved. The code performing simulations is supplied, and works properly with OpenBUGS. The code itself could be a bit more commented, as currently it requires careful examination with parallel reading the ms and there are things that one has to deduce.

Experimental design

Research question could be stated more explicitly. The 2nd sentence in the Background section could be replaced with clearly formulated question. Methods are described sufficiently, though I suggested few corrections. Conclusions are sound and do not exceed the reported findings (see General comments).

Validity of the findings

I believe the main finding is of importance: that the strength of density regulation amplifies fluctuations due to environment will help to better describe population dynamics and to answer new questions. This gives space for new, intriguing research in the field.

Additional comments

This includes detailed comments.

Abstract
L19-20, Background: research question could be stated explicitly and replaced with the second sentence which is just saying the issue Authors were working on has not received enough attention. Which is not a research question.
L38-41, Discussion: First sentence is very long and is better divided into two (ending after “environments”). I actually also believe that at the end of this section, Authors could stress the importance of the main finding more - adding that separating these processes will help improve population dynamics modelling would not be overstatement.

Introduction
L95-97: it would be better to say if any attempts have been made at all to separate these two.
L108: better say “other” instead of “various” (but just Ricker is considered)

Material & Methods
Could be divided into two main parts: Analytical decomposition... and Simulation study, which could improve clarity. I am not sure where the ‘decomposition’ part belongs to as currently you are reporting just simulations in Results section. But the ‘decomposition’ part is also your result, no less important.
L124: “is” missing?
Eq (7): so long run mean is determined by carrying capacity? Worth discussing.
L197: the variance due to density regulation should be better defined here, difficult to read. Just place the notation after the definition?
L205: calling beta density-dependence effect should appear well before, when it is first introduced
L213-219: autoregressive structures can be incorporated also in standard package models, making CI’s valid I think. What Dennis & Taper 1994 conclude about their discussion – do these methods provide valid CI’s? And, if Bayesian solutions are similar, does it mean they also provide not valid CI’s? is this the reason you do not report individual estimates of beta (along with their CI’s), but just a (kind of) summary using boxplots on the graps? If CI’s from Bayesian models you use provide correct CI’s, consider adding a sort of ‘coverage’ statistics saying how often, from a single simulation run, these CI’s include true (simulated) value (I mean proportion of variance due to density regulation, Fig. 2).
Line 216: “mot”, a typo
Line 221-230: perhaps make it a separate section (Ricker model)
Line 232: this is where numbering of sections appears unexpectedly
Line 237-239: I realize the work is theoretical, but what if these parameters change?
Line 263-267: long sentence, difficult to follow, please rewrite

Results
Line 277-278: formal, but are there any slopes (plural)? Or, is there a single slope? I mean there could be, and it surely becomes steeper for higher beta, but you just take three fixed k’s (or betas) in simulations. If you need slope to further support your conclusions (I think you don’t), vary k in simulations along a sensible range, compute and draw the slopes, or rewrite.
Line 284: ref to Supplemental Fig. S1. Simple histograms could be replaced by nicer violin-type plots, where one can show both the individual results (estimates of env variance) and the distribution (e.g., six violins in a horizontal setting, with colour denoting Gompertz and Ricker). My suggestion is to make Y axis range wider, say, 0.5-1.5 or even 0-2.
L287-290: repeated from Methods, nothing new. Delete?
L291: word ‘variance’ missing? Would improve clarity if followed by explanation like “... stationary variance (ie, var due to density regulation) increased...”
L292: perhaps delete “However, ” from here – wasn’t it (approximately) the same for Ricker model as well? It seems so as judged from figure. “However” indirectly suggests it was not the case with Ricker. Actually, both models should be mentioned, not just Gompertz.
L300-304: this looks more like a Discussion part (you say you expect something; this is not the result).

Discussion
L338: ‘steeper slopes’ again

Figures
Figure 1
(A). I miss information for what k is this. If it’s part of the same data as in (B), then something is wrong. At the one hand frequencies sum to >>50, so more than one simulation, but the values clearly do not include the ones for beta = 0.75 on (B) as the median there is well above 2.0. So please be clear what exactly is depicted on (A).
(B) Even if these boxlots are drawn with the default setting, it is nice to explain that there is the median (?) inside the box, and what range box and whiskers show. Note one can modify standard settings. Weren’t there any outlying values or were they removed? Actually I believe a violin-type plot would be more informative here, since you can show both individual estimates and the distribution.
Figure 2
Both panels could be replaced by a single one, a violin-type, with added theoretical values from eqn 8 for each beta to clearly show the match between simulated and theoretical proportions. My suggestion regarding boxplots description as for Figure 1.

·

Basic reporting

The biological context could be better explored

Experimental design

OK

Validity of the findings

Some model assumptions are not realistic

Additional comments

My review was submitted in a separate file.

Reviewer 3 ·

Basic reporting

Submited paper is well writen. The structure is clear. Few small typos should be corrected.

Experimental design

The paper is purely theoretical and contains novel method of analysis of population growth models with stochastic component describing environmetal uncertainity. The analytical results are tested by monte carlo simulations. The method of reduction of the initial model to the autoregression model is valuable contribution an it shows how random effect from the previous time moment affected by current random perturbation determine the variance of fluctuations of the population size at equilibrium. this technique is simple but potentially useful. However, paper contains serious disadvantages which are not faults of the authors, but are inherited from errors on previous works which are not succesfully resolved but unfortunately still very popular. The next section contains more detailed questions:

Validity of the findings

Authors use classical (and very old) Gompertz and Ricker models. The problem is that both those models contain a flaw inherited from the Verhulst's logistic model. They use the phenomenological growth rates and the malthusian parameter r is multiplied by supression term. This type of models was designed to mimick the sigmoid patterns observed in the ecological data, however their parameters, especially the Carrying capacity, describing the arbitrary equilibrium are problematic from the mechanistic point of view. The problem is following: if we consider that the malthusian growth rate is related to the balance between fertility and mortality (even if we do not assume that we can prove by first order Taylor expansion that Malthusian growth rate is proportional to per capita (number of births)-(number of deaths)). Then multiplication by suppression term (1-n/K) leads to the suppression birth rate and death rate simultaneously. In effect we obtain the immortal population at equilibrium. another problem resulting from this approach is so called Levins Paradox. For r<0 and n0>K continuous model escapes to infinity. Thus, for declining populations at high densities, suppression term induces the exponential explosion. This and other problems resulting from this form of growth suppression are discussed in (Kuno 1991). Simple solution of this problem based on assumption that only birth rate should be suppressed was proposed in Kozłowski 1980 and in the classical book of Łomnicki 1988. Later, this aspect was discussed in the debate on the logistic model Ginzburg (1992) Gabriel et al. (2005) finished by clarification of the concept of carrying capacity as the maximal population load (Hui 2006, 2015). Resulting latest development is focused on fully mechanistic reductionist models assuming that the population growth is limited by some limiting factor, such as availability of nest sites for newborns (Argasinski and Rudnicki 2017). In those models equilibrium is defined as the balance between all fertility and mortality factors instead arbitrary carrying capacity. This resembles the concept of emergent carrying capacity, that can be found in epidemic models (Bowers et al., 2003; Sieber et al., 2014). it will be interesting how the methodology proposed by authors will work, when applied to more mechanistic models. Specific comments:

1. Introduction:
Authors present the problem of density dependence an support the introduction with respective references. However we have two issues here:
-one paragraph is related to Alle effects, while results from the paper seem to be not related to thos problem at all. More details will be helpful
-distinction between density dependent and density independent mechanisms is introduced. this problem is essential in the emerging eco-evolutionary synthesis (Post and Palcovacs 2009, Schoener 2011,Hendry 2016) and the interplay between frequency dependent selection and growth suppression leading to the eco-evolutionary feedback can be found in Argasinski and Broom (2018).

2. Equation 1:
Potential source of the problems is the fact that the noise term is completely independent from the deterministic growth part. Thus, it is possible that small growth, for small r, is affected by strong noise. In mentioned above more mechanistic models we can imagine that the random noise may result from the finiteness of the population such as the case when relatively low number of newborns investigate nestsites to find a free place. Finite number of trials will lead to binomial noise term, which can be approximated by Gaussian distribution. However, in this case the noise term will depend on the population size and the probability of successful recruitment of newborn that found free nest site. This is hypothetical case to show that the approach used in eq.1 is somewhat simplistic. In addition the term "environmental stochasticity" should be explained in greater detail. Why it affects growth rate in this way?

3. eq's (2) to (6)
Why authors ignore specific cases leading to strange behaviour such as beta equal to zero for r=k. Why beta is limited to -1? For r>k we will have smaller values. By the way, these effects show the absurd of the arbitrary equilibrium k of the classical models.

4. eq (8)
This is one of the main results. For me the unusual amplification of the variance at equilibrium seems to be another artifact, similar to the Levins paradox, produced by classical models. Beta is close to 1 for small r and very big k. Huge noise for slowly growing populations approaching very high densities is very strange.

5. line 191-192
For me density independent model is the free exponential growth. if we have white noise oscilations around r, then we have very strong growth suppression. In addition, equilibrium population size equal to the per capita growth rate is very bizarre. So, definitely something is wrong here..
What means the statement that "beta is not statistically different from zero"? This is a constant, so it can be simply zero. Randomness in in the Gaussian term.

6. line 216 and 218
typos: mot, mon-informatives

7. lines 344-248
This paragraph is vaugue. What means "overlooks the stochastic component etc."

Additional comments

Authors should be more critical to the obtained behaviour of the model. Some aspects seems to be obvious artifacts resulting from the flaws of the classical models. To be honest, i am not a big fan of the classic models and results from the submited paper convinced me that they are complete junk. However, the proposed method should be very useful, when applied to some mechanistic model. It is interesting how the proposed method will work in the modern model with explicit distinction between suppressed birth rate and constant death rate.

references:


Argasinski, K., & Broom, M. (2018). Evolutionary stability under limited population growth: Eco-evolutionary feedbacks and replicator dynamics. Ecological Complexity, 34, 198-212.

Argasinski, K., & Rudnicki, R. (2017). Nest site lottery revisited: Towards a mechanistic model of population growth suppressed by the availability of nest sites. Journal of Theoretical Biology, 420, 279-289.

Bowers, R.G., White, A., Boots, M., Geritz, S.A.H., Kisdi, E., 2003. Evolutionary
branching/speciation: contrasting results from systems with explicit or emergent
carrying capacities. Evol. Ecol. Res. 5, 883–891.

Hendry, A. P. (2016). Eco-evolutionary dynamics. Princeton university press.

Sieber, M., Malchow, H., Hilker, F.M., 2014. Disease-induced modification of prey
competition in eco-epidemiological models. Ecol. Complex. 18, 74–82.

Post, D. M., & Palkovacs, E. P. (2009). Eco-evolutionary feedbacks in community and ecosystem ecology: interactions between the ecological theatre and the evolutionary play. Philosophical Transactions of the Royal Society B: Biological Sciences, 364(1523), 1629-1640.

Gabriel, J.P., Saucy, F., Bersier, L.F., 2005. Paradoxes in the logistic equation? Ecol.
Model. 185, 147–151.

Ginzburg, L.R., 1992. Evolutionary consequences of basic growth equations. Trends Ecol.
E 7, 133.

Hui, C., 2006. Carrying capacity, population equilibrium, and environment's maximal
load. Ecol. Model. 192, 317–320.

Hui, C., 2015. International Encyclopedia of the Social & Behavioral Sciences. volume 3.
Elsevier.

Łomnicki, A., 1988. Population Ecology of Individuals. Princeton University Press.

Kozłowski, J., 1980. Density dependence, the logistic equation, and r- and K-selection: a
critique and an alternative approach. Evol. Theor. 5, 89–101.

Schoener, T. W. (2011). The newest synthesis: understanding the interplay of evolutionary and ecological dynamics. science, 331(6016), 426-429.

---

## Round 0.2 · Major Revisions

The reviewers have provided their second in-depth reviews and although they acknowledge changes introduced, they still rise serious issues. I would like to invite the Authors to carefully apply those, especially:

- to reconsider comments raised by Reviewer 3 (as they rightfully recognise - most of original comments where not taken into account, a better justification and/or reconsideration of this decision should be provided)
- to clarify the "density dependence" parameter: both reviewers bring this to light and note that this name is probably not the best descriptor of what this parameter really means
- to improve on the biological reality of simulations (e.g., by addressing reviewer comments pointing out biologically unrealistic scenarios arising in the model, and supplementing the paper with sensitivity analyses that demonstrate the robustness of the findings)

·

Basic reporting

See attachment

Experimental design

See attachment

Validity of the findings

See attachment

Additional comments

See attachment

Reviewer 3 ·

Basic reporting

It is a bit surprising that Authors ignored nearly all my previous comments or replied for them in the very vaugue way. The relation of the results from the paper to new (and old but forgotten) research suggested in my previous review is not even mentioned in the new version of the paper and in the reply. The mentioned paper of Geritz and Kisdi is related to the model from the paper only in the limited way. However, some assumptions presented there may be adopted. My main concern, which is still critical issue can be formulated in a simple way. Authors start from eq.1 which follows

Y_t=Y_t-1{exp(r(1-ln(Y_t)/ln(K)))+epsilon}


Another thing. This model can be mechanistically interpreted as the seasonal model when all adults die (non overlapping populations) and are replaced by recruits and their survival is described by the bracket. Similarly to the original Ricker's setup for salmons as here (see the number of specific assumtions there):

http://www.mat.unimi.it/users/naldi/Ricker_model.pdf

or similarly to the latest nest site lottery models (then K is the maximal population load) mentioned in previous review. Then we have raw growth rate r and multiplicative suppression bracket describing the impact of density dependece.
The question is does the results hold for overlapping populations?

But let usfocus now on the main issue. Thus population without impact of the density dependence is described by exponential growth with rate r. Density dependence acts as the juvenile recruitment survival described by specific brackets for each model. Therefore unsuppressed version of eq (1) will be

Y_t=Y_t-1{exp(r)+epsilon}

suppressed form can be described in the from

Y_t=Y_t-1{exp(r(1-ln(Y_t)/ln(K)))+epsilon}
= Y_t-1{exp(r-r*ln(Y_t)/ln(K))+epsilon}

and -r*ln(Y_t)/ln(K) is the factor responsible for density dependence. when it is equal to zero we have unsupressed growth (such as unlimited number of nest sites in the nest site lottery models)


Therefore formula beta which links intrinsic growth rate with k is badly interpreted as the measure of deensity dependece. This interpretation is wrong, which is shown by this white noise case, Unsuppressed popoulation of salmons will not randomly fluctuate around a number of newborns of a single adult salmon. This is biological nonsense. For lack of density dependence beta=1 for beta=0 we have r=k which is extremely strong density dependence (thus lines 229-230 are completely wrong). Thus all interpretations of the formula for variance are probably exactly opposite to the statements in the discussion. This is the critical bug and should be fixed. This issue emphasizes the importance of clear mechanistic interpretation of the parameters of the models and the advantage of the mechanistic models mentioned in the previous review.


Then the question arise what is additive environmental noise? For huge variances of this factor we can obtain negative population sizes thus it is problematic. In the case of Ricker's salmons the environmental noise will affect for example the abundance of predators, thus it will be part of K. Similarly for nest site models it can describe fluctuations in the availability of free nest sites. What it is n the model (1) from the paper? In the model (1) the noise affects the per capita number of sucessfully recruited newborns thus it is density independent. How it can be mechanistically interpreted?

Experimental design

no comment

Validity of the findings

no comment

Additional comments

no comment

---

## Round 0.3 · accepted · Accept

The Academic Editor is no longer available and so I am issuing a decision in my capacity as Section Editor.

Thank you to the reviewers for extensive and helpful comments that have improved this manuscript and that have made it acceptable for publication in PeerJ. Thank you also to the co-author for their work on improving the manuscript and for contributing to PeerJ and the wider community. Please note the final very minor comments from Rev. 3 during the production/galley process.

Reviewer 3 ·

Basic reporting

In the latest version, the main critical bug which is wrong intepretation of the parameter beta is fixed and the paper reached maturity and can be published. The very last thing: in line 246:

When β=0, which corresponds to r=k in the standard Gompertz model formulation, the population trajectory is follows a Gaussian a white noise process shifted atshifted at r

fragment "which corresponds to r=k in the standard Gompertz model formulation," is removed. I think it will be good to restore it since it emphasizes that this is extreme case when carrying capacity is extremely low.

Experimental design

no comment

Validity of the findings

no comment

Additional comments

no comment